# Combination of *Saffron* (*Crocus sativus*), *Elderberry* (*Sambucus nigra* L.) and *Melilotus officinalis* Protects ARPE-19 Cells from Oxidative Stress

**DOI:** 10.3390/ijms26041496

**Published:** 2025-02-11

**Authors:** Alessandra Puddu, Massimo Nicolò, Davide C. Maggi

**Affiliations:** 1Department of Internal Medicine and Medical Specialties, University of Genoa, 16132 Genoa, Italy; davide.maggi@unige.it; 2Department of Neuroscience, Ophthalmology and Genetics, University of Genoa, Viale Benedetto, 16132 Genova, Italy; massimo.nicolo@unige.it; 3Fondazione per la Macula Onlus-Genova, Piazza della Vittoria, 16121 Genova, Italy

**Keywords:** retinal pigment epithelium, *Saffron*, *Elderberry*, *Melilotus*, oxidative stress, retinal diseases, inflammation

## Abstract

Oxidative stress is considered a common underlying mechanism in many retinal degenerative diseases and is often associated with inflammation. The use of dietary supplements containing *Saffron* has beneficial effects in ocular diseases, though the molecular mechanisms are still unclear. In this study, we investigated how *Saffron* can exert protective effects against oxidative damage in retinal pigment epithelial cells (ARPE-19) and whether its combination with *Elderberry* and *Melilotus* may have additive beneficial effects. ARPE-19 cells were pretreated with *Saffron* alone or in a mix containing Saffron, *Elderberry* and *Melilotus*, then exposed to hydrogen peroxide (H_2_O_2_) for 3 h. Afterwards, we evaluated cell viability, oxidative stress and inflammatory status. Our results showed that H_2_O_2_ reduced cell viability and total glutathione levels, while increasing caspase-3, caspase-1 and LDH activity. Moreover, H_2_O_2_ triggered ROS production, glutathione oxidation and IL-1β secretion. Pretreatments with *Saffron* alone or with the mix counteract these damaging effects by improving cell viability, reducing oxidative stress and enhancing SOD2 expression. Pretreatment with the mix activated the NRF2 pathway and was more effective than *Saffron* alone in preventing caspase-1 activation. These findings suggest that the combination of *Saffron*, *Elderberry* and *Melilotus* could have therapeutic potential in the prevention and treatment of retinal degenerative diseases.

## 1. Introduction

The retina is one of the tissues with a higher oxygen consumption rate, thus implying also a higher susceptibility to oxidative stress, which is caused by the imbalance between reactive species production and the ability to scavenger them [1,2]. In addition to oxidative stress resulting from oxygen metabolism, retinal oxidative stress is also associated with the production of Reactive Oxygen Species (ROS) generated by the abundant presence of poly-unsaturated fatty acids (PUFAs) in the outer membrane of the photoreceptors, as well as by the continuous ROS production caused by light exposure [3,4,5]. Consequently, ROS, such as hydrogen peroxide (H_2_O_2_), superoxide and hydroxyl anions, accumulate with aging and their production is further accelerated by hyperglycemia [4,6]. Oxidative stress plays a crucial role in the development of several ocular diseases, including age related macular degeneration (AMD) and diabetic retinopathy (RD) [4,5]. Indeed, the increased production and accumulation of ROS alters key cellular functions, including metabolism, by increasing the polyol and the hexosamine pathways, and affects the activation of protein kinase C isoforms [4]. Furthermore, ROS promote the accumulation of advanced glycation end products (AGEs), which, in turn, stimulates the production of more ROS, establishing a vicious cycle [7,8]. In addition, epigenetic modifications induced by ROS lead to decreased expression of antioxidant defenses, thus raising potential damage due to oxidative stress [4,5]. This vulnerability condition is further worsened by inflammatory milieu generated by activation of resident retinal cells or by infiltrating inflammatory cells [9]. Indeed, oxidative stress impairs retinal cell function, leading to the secretion of pro-inflammatory factors, such as Monocyte Chemoattractant Protein-1 (MCP-1), interleukin (IL)-1β and IL-18. These factors trigger several intracellular signaling pathways that stimulate the production of additional cytokines and chemokines [9]. This, in turn, results in an inflammatory environment that activates inflammatory cells, which contribute to exacerbation of the inflammatory process, thus creating a vicious cycle that increases both inflammation and oxidative damages to retinal cells [10,11]. Therefore, although several factors contribute to eye diseases, oxidative stress may be considered as a common mechanism at the basis of many degenerative retinal conditions.

Among retinal cells, the retinal pigment epithelial (RPE) cells play a main role in maintaining retinal homeostasis [12]. Indeed, due to their location, activities of RPE cells may affect the Bruch’s membrane and the choroid at the basolateral side, and the neural retina at the apical side. Moreover, RPE cells are also a source of pro-inflammatory cytokines, thus acting as one of the main actors in the inflammatory process [13]. Consequently, RPE dysfunction is associated with several ocular diseases that may lead to visual loss, such as AMD and DR [14,15]. Therefore, preserving the function of RPE cells is of particular importance in preventing the onset and progression of retinal pathologies.

Several herbal drugs and their extracts are known to have antioxidant proprieties, and are the objects of a great deal of research [16,17]. The use of natural compounds in topical preparations and as dietary supplements may be a beneficial alternative to pharmacological treatments to control symptoms and to prevent or slow the progression of ocular diseases [18,19,20]. Recently, a dietary approach has been proposed in the form of a food pyramid that promotes the intake of micro-greens and spices, such as Saffron and Curcumin, 3–4 times for week to prevent or support the treatment of DR, AMD, and cataracts [21].

*Saffron* (*Crocus Sativus*) is one of the most widely characterized natural compounds with anti-inflammatory proprieties [22]. It is an annual species mainly produced in Iran and in Mediterranean countries, traditionally used as a food additive and aromatic spice [23]. The therapeutic activities of Saffron are due to its main bioactive components, crocetin, a carotenoid compound, and crocin, which derives from crocetin hydrolyzation during intestinal absorption [24,25]. Several studies have demonstrated that oral supplementation with *Saffron* has beneficial effects in ocular diseases, such as DR, AMD, retinitis pigmentosa and glaucoma [22,26]. However, there is an increasing interest in the association of different antioxidant compounds with the aim of improving the beneficial effects of a single treatment [17]. For instance, recent evidence showed that Resvega^®^, a nutraceutical formulation composed of omega-3 fatty acids and resveratrol, has beneficial effects in preventing oxidative damage and inflammatory response in RPE cells [27,28,29]. More recently, a new nutraceutical preparation, Visuretin^®^, has been formulated that combines Saffron with *Elderberry* (*Sambucus nigra* L.) and *Melilotus* (*Melilotus officinalis*).

*Elderberry* is a species complex of flowering plants in the family Adoxaceae with many health functions [30]. Both flowers and fruits contain several bioactive compounds, including phenolic acids, flavanones, flavonols, and anthocyanins, responsible for anti-inflammatory and antioxidant properties [31,32,33,34]. *Melilotus officinalis* belongs to the family Fabaceae and has been used as an herbal medicine from ancient times [35,36]. Its beneficial effects are mainly due to the biological activity of the coumarins formed during the drying process [37,38]. Despite encouraging results in the literature, the mechanisms of action by which these natural compounds provide beneficial effects in RPE cells are not yet completely understood.

Our study aims to characterize the molecular mechanisms through which *Saffron* may exert protective effects against oxidative damage in RPE cells, and whether its association with *Elderberry* and *Melilotus* may exhibit significantly additive antioxidant effects.

## 2. Results

### 2.1. Saffron and SEM Mix Protects RPE Viability Against Oxidative Stress

Firstly, we tested the cytotoxic effects of different concentrations of hydrogen peroxide (H_2_O_2_) ranging from 50 to 500 µmol/L on ARPE-19 cells. As shown in Figure 1A, only a greater concentration of H_2_O_2_ significantly reduced the viability of ARPE-19 cells. Therefore, 500 µmol/L H_2_O_2_ has been used in further analysis. *Saffron* (SAF) was used at a concentration of 40 µg/mL, as determined by findings from previous studies [39,40]. Due to the limited data in the literature about the use of *Elderberry* (E) and *Melilotus* (M), we investigated their protective effects on the viability of ARPE-19 cells using concentrations ranging from 50 to 200 µg/mL. The results showed that only the highest concentration (200 μg/mL) of both compounds significantly improved cell viability when cells are exposed to 500 µmol/L H_2_O_2_ (Figure 1B,C). Based on these findings, we used a mix composed of 40 μg/mL SAF + 200 μg/mL E + 200 μg/mL M (SEM mix) for the subsequent experiments.

Addition of *Saffron* (SAF), as well SEM mix, to the culture medium significantly improved cell viability when cells were exposed to H_2_O_2_ (Figure 2A). To characterize ARPE-19 cell death induced by H_2_O_2,_ we investigated the activity of caspase-3 and LDH. Caspase-3 is considered as a key effector enzyme in inducing apoptotic cell death [41], whereas lactate dehydrogenase (LDH) is widely used as a marker of necrosis, because it is a stable cytosolic enzyme, which is rapidly released into the cell culture medium upon disruption of the plasma membrane [42]. Here, we found that caspase-3 activity was slightly increased by treatment with H_2_O_2_ (Figure 2B). Pretreatment with SEM mix significantly reduced activation of caspase-3 even in the presence of H_2_O_2_. The activity of LDH was significantly increased by H_2_O_2_ (Figure 2C). Pretreatment with SAF did not affect LDH activity, whereas the SEM mix significantly reduced the release of LDH in comparison to CTR. In addition, the SEM mix prevented the increased activity of LDH induced by H_2_O_2_ (Figure 2C).

### 2.2. Saffron and SEM Mix Reduced Oxidative Stress

To evaluate whether the cytoprotective effects of SAF and SEM mix on ARPE-19 cells exposed to 500 µmol/L H_2_O_2_ were related to their antioxidant proprieties, we evaluated oxidative stress by measuring the levels of ROS in cell culture medium. Exposure of ARPE-19 cells to H_2_O_2_ induced a significant increment in ROS production in comparison to cells incubated with control medium (Figure 3). Pretreatment with SAF or SEM mix significantly decreased H_2_O_2_-induced ROS production. The results indicated that SAF had a stronger effect than the SEM mix in reducing ROS production.

Glutathione (GSH) is an abundant antioxidant capable of preventing damage caused by ROS by maintaining redox balance [43,44,45]. In the presence of oxidative stress, GSH is oxidized and forms a dimer connected by a disulfide bond (GSSG). Therefore, the ratio of reduced to oxidized glutathione (GSH/GSSG) is an indicator of cell health and oxidative stress. As shown in Figure 4A, exposure of ARPE-19 cells to H_2_O_2_ significantly reduced total GSH (Figure 4A) and increased its oxidation (Figure 4B), thus resulting in a decreased GSH/GSSG ratio (Figure 4C). Pretreatment with SAF or with the SEM mix counteracted the formation of GSSG (Figure 4B). However, the ratio between GSH to GSSG was significantly improved only when ARPE-19 cells exposed to H_2_O_2_ were pretreated with the SEM mix (Figure 4C).

### 2.3. SEM Mix Activates the NRF2 Pathway

To explore the molecular mechanisms underlying the protective effects of SAF and SEM mix, we investigated whether these treatments reduced oxidative stress by modulating the expression of antioxidant protective factors. First, we assessed the expression of Nuclear Factor Erythroid 2-Related Factor 2 (NRF2), a key transcription factor that regulates the expression of various antioxidant enzymes [46]. In response to oxidative stress, NRF2 dissociates from Keap1 and translocates into the nucleus, where it binds to antioxidant response elements (AREs) in the promoter region of several antioxidant genes, inducing their transcription [46]. Our results showed that the protein expression of NRF2 is not significantly affected by incubation with 500 µmol/L H_2_O_2_ or by pretreatment with SAF (Figure 5A). However, expression of NRF2 was significantly up-regulated whenARPE-19 cells were pretreated with the SEM mix. This increased expression of NRF2 induced by the SEM mix was maintained even after exposure to 500 µmol/L H_2_O_2_. Then we evaluated the protein levels of Keap1, but we did not observe any statistically significant differences among the experimental conditions (Figure 5B). Finally, we analyzed the expression of Superoxide Dismutase 2 (SOD2), a gene induced by NRF2 [47]. Protein expression of SOD2 was significantly increased in ARPE-19 cells pretreated with SEM mix, even after exposure to 500 µmol/L H_2_O_2_ (Figure 5C).

### 2.4. Saffron and SEM Mix Reduced Caspase-1 Activity

Then, we verified whether SAF and SEM mix may activate anti-inflammatory pathways in ARPE-19 cells exposed to oxidative stress. Caspase-1 is activated after assembly of the inflammasome complex and is involved in inflammasome mediated cell death in retinal degenerations [48,49]. As shown in Figure 6, we found that H_2_O_2_ significantly increased caspase-1 activity in ARPE-19 cells. Treatment with SAF did not affect the activity of caspase-1, but restored the increment, due to H_2_O_2_ exposure, to the level of untreated cells. The SEM mix was able to significantly decrease both constitutive and H_2_O_2_-induced caspase-1 activity. Furthermore, the SEM mix was more effective than SAF alone in reducing H_2_O_2_-induced caspase-1 activity.

Once activated, caspase-1 plays a key role in inflammation by cleaving the precursor form of several cytokines into their active forms, which are released by the cells [48]. Among the key substrates of caspase-1, Interleukin-1β (IL-1β) is considered one of the major pro-inflammatory mediators [50]. To verify whether the decrease in caspase-1 activity is linked to a reduction in pro-inflammatory cytokine secretion, we measured the amount of IL-1β in the culture medium of ARPE-19 cells. The results showed that ARPE-19 cells cultured under standard conditions secrete a low amount of IL-1 β (1.096 ± 0.08 pg/mg protein) (Figure 7), which was further reduced by culture with SAF (0.96 ± 0.13 pg/mg protein) and the SEM mix (0.84 ± 0.07 pg/mg protein), although the reduction was not statistically significant. Exposure to 500 µmol/L H_2_O_2_ significantly increased IL-1 β release. This increment was completely prevented when cells were pre-treated with SAF or the SEM mix for 24 h.

### 2.5. Saffron and SEM Mix Improve Cell Morphology

Finally, we documented the morphological characteristics of ARPE-19 cells under each experimental condition using light microscopy. The images in Figure 8 show that exposure of ARPE-19 cells to H_2_O_2_ reduced the cell volume and altered the cell shape. Treatment with SAF partially improved cell shape, but the cell volume remained reduced, whereas SEM pretreatment fully preserved the morphology of ARPE-19 cells, even in the presence of H_2_O_2_.

## 3. Discussion

Oxidative stress is considered a common mechanism that leads to the development and progression of many retinal degenerative diseases [9]. The role of dietary antioxidants and their potential beneficial therapeutic effects elicit great interest among researchers, especially because dietary supplements represent a simple and cost-effect strategy to treat retinal degenerative diseases. In this work, we aim to characterize the mechanisms through which *Saffron* can exert protective effects against oxidative damage in RPE cells and whether its association with *Elderberry* and *Melilotus* may further increase RPE defense.

In this study, we found that hydrogen peroxide strongly increased LDH release almost without affecting the activity of caspase-3, suggesting that the reduced viability of RPE cells is mainly due to necrosis. This agrees with the knowledge that a strong oxidative stimulus results in increased permeability of the plasma membrane [51]. Interestingly, the SEM mix was able to reduce the constitutive activity of both caspase-3 and LDH, suggesting a greater potential in counteracting death insults in comparison to *Saffron* alone.

Oxidative damage starts with ROS production, which, once generated, may damage other cellular components, such as lipids, DNA and proteins, thus amplifying the detrimental effects of ROS [52]. RPE cells contain many endogenous systems of antioxidant defense which act synergically to neutralize ROS, thus maintaining redox homeostasis [53]. To evaluate whether cyto between oxidative and reduction processes cannot be maintained by endogenous antioxidant defenses, it is necessary to help cells via therapeutic antioxidant strategies. An imbalance between generation of reactive ROS and antioxidant defense systems represents the primary cause of cell dysfunction [54]. Antioxidant agents may act at different levels with broad mechanisms, which range from disrupting the free radical chain reactions to regulating the activity of enzymatic agents, such as superoxide dismutase and glutathione peroxidase [52]. Therefore, treatment with antioxidants may be designed to neutralize free radicals and increase the antioxidant defense system. The antioxidant proprieties of *Saffron* arise from both its direct action as a free radical scavenger, and its regulation of genes involved in the antioxidant response [22,25,55]. Indeed, *Saffron* has been shown to activate multiple pathways, including the inhibition of free radical production, reduction of lipid peroxidation and an increase in the levels and activity of antioxidant enzymes. Recently Di Marco et al. demonstrated that *Saffron* administration reduced retinal degeneration in patients with AMD, suggesting that *Saffron* may have potential added benefits compared to traditional antioxidant treatments [56]. In line with these findings, we found that *Saffron* alone or in combination with Elderberry and Melilotus was able to reduce ROS production. It is well known that RPE cells, due to their high rate of oxygen (O_2_) consumption, developed an efficient antioxidant machinery to counteract ROS production during metabolic processes [57,58]. Among the key antioxidant enzymes, Superoxide dismutase (SOD) and Catalase (CAT) play essential roles in protecting against superoxide radicals: SOD neutralizes superoxide anion, converting it to hydrogen peroxide (H_2_O_2_) and O_2_, then CAT completes the detoxification by reducing H_2_O_2_ into water and oxygen [59,60]. Interestingly, the system used to evaluate ROS production measures the level of hydrogen peroxide formed in cell culture, thus it indirectly indicates the capacity of the cells to eliminate H_2_O_2_, reflecting the activity of both SOD and CAT. Therefore, our findings suggest that both *Saffron* and SEM mix reduced ROS production by stimulating the activity of CAT. In addition, our results demonstrated that SAF and SEM mix affect the redox balance by regulating the ratio between reduced to oxidized forms of glutathione. Considering that the nutraceuticals used in this study prevented the rise of GSSG in the presence of hydrogen peroxide, without increasing the amount of GSH, these results suggest that cells pretreated with SAF or SEM mix need to oxidize a lesser amount of GSH to counteract oxidative stress. Moreover, they prevented the rise in the formation of the oxidized form of glutathione, meaning that treatment with these nutraceuticals may enhance endogenous antioxidant defenses, in addition to glutathione, in scavenging ROS, confirming that the addition of these nutraceuticals results in lower production of ROS in RPE cells. The increased expression of SOD2 observed when ARPE-19 cells are pre-treated with SAF or SEM mix may also contribute to reducing H_2_O_2_ and ROS formation, when cells are exposed to oxidative stress. Another mechanism potentially involved in the antioxidant response of ARPE-19 cells is the activation of the NRF2 pathway. The transcriptional activity of NRF2 plays a key role in helping RPE cells to counteract oxidative stress. However, the ability of NRF2 to translocate into the nucleus depends on its interaction with Keap1. Our findings showed that Keap1 expression remained at the level of control cells in all experimental conditions. However, the expression of NRF2 increased when cells were cultured with SEM mix, leading to the prevalence of NRF2 over Keap1 expression. Therefore, these results suggest that pretreatment with SEM mix allows the establishment of conditions that favor nuclear translocation of NRF2. This hypothesis is further supported by the increased expression of SOD2, even when cells are exposed to oxidative stress.

Inflammation is considered a key pathogenic factor in several degenerative diseases of the retina [48]. Our results suggest that the protective effects of *Saffron* and SEM mix against oxidative stress may also prevent the activation of the pro-inflammatory pathway in RPE cells. To test this hypothesis, we evaluated the activity of caspase-1, one of the component of the inflammasome, and the release of IL-1β [48,49]. Caspase-1 activity is of particular importance because it results in the cleavage of pro-IL-1β and pro-IL-18 [48], thus leading to the increased release of these pro-inflammatory cytokines and the onset of the inflammatory milieu. Although recent studies have reported that *Saffron* suppresses inflammasome activation [61,62,63], we found that *Saffron* was not able to counteract activation of caspase-1 induced by oxidative stress in our model. Probably, this discrepancy may be due to differences in experimental conditions. Considering that SEM mixture decreased both constitutive and H_2_O_2_-induced caspase-1 activity, our findings suggest that *Elderberry* and/or *Melilotus* may be responsible of this beneficial effect. As expected, the decrease in caspase-1 activity was associated with reduced IL-1β secretion in cells pretreated with the SEM mix. Interestingly, our results showed that pretreatment with SAF alone prevented the rise in IL-1β secretion induced by oxidative stress. These findings suggest that the improvement in redox balance achieved with the addition of SAF and SEM mix may contribute to the prevention of the inflammatory response. In addition, the SEM mix was more efficient than *Saffron* alone in increasing the ratio of GSH to GSSG; therefore, it can be hypothesized that a stronger inhibition of ROS production, likely due to SEM mix, may prevent the activation of the inflammasome complex, and that the combination of Saffron with *Elderberry* and *Melilotus* may result in synergic effects. Overall, the morphological aspects of ARPE-19 cells confirm the greater beneficial effects of pre-treatment with SEM mix in comparison to *Saffron* alone. Indeed, despite similar levels of cell proliferation, pretreatment with SEM mix allowed for better recovery of cells exposed to oxidative stress than *Saffron* alone.

In conclusion, our results demonstrate that both Saffron alone and its combination with *Elderberry* and *Melilotus* are able to counteract the damaging effects of hydrogen peroxide. However, the mix is more effective in preventing inflammation, suggesting that the combination of different bioactive compounds may give stronger antioxidant protection. These findings provide evidence that adding Saffron, *Elderberry* and *Melilotus* to the culture medium confers cytoprotective effects on RPE cells. In particular, this combination activates both antioxidant and anti-inflammatory pathways, restoring the redox balance and preserving cell morphology. Overall, our results suggest that drug formulations containing a combination of Saffron, *Elderberry* and *Melilotus*, whether developed for topical use or as dietary supplement, may offer potential benefits in the prevention and treatment of retinal degenerative diseases.

## 4. Materials and Methods

### 4.1. Cell Culture and Experimental Conditions

The human retinal pigment epithelial cell line ARPE-19 was obtained from the American Type Culture Collection (American Type Culture Collection, Manassas, VA, USA) and cultured as previously described [64]. Cells were grown in standard medium [DMEM/F12 1:1 medium (Life Technologies Italia, Milan, Italy) supplemented with 10% fetal bovine serum and 2 mmol/L glutamine (Euroclone, Milan, Italy)] for 4 days and exposed for 3 h to 0, 50, 100, 200 and 500 µmol/L hydrogen peroxide (H_2_O_2,_ Euroclone, Milan, Italy). The concentration 500 µmol/L H_2_O_2_ was selected as that which reduced cell viability. Afterwards, cells were pretreated for 24 h with 40 μg/mL of *Saffron* (*Crocus sativus* L.) standardized to contain 3% crocins (Affroneye, Pharmactive Biotech Products, Alcobendas, Madrid, Spain) (SAF) or various doses (0, 50, 100 and 200 µg/mL) of the high-quality elderberry plant extract Eldosamb^®^ (*Sambucus nigra* L., Anklam Extrakt GmbH, Anklam, Germany) (E) or of *Melilotus officinalis* containing 12% coumarin (Nutraceutica srl, Monterenzio, Bologna, Italy) (M) (all the compounds were kindly gifted by VISUfarma spa, Roma, Italy). Then ARPE-19 cells were exposed to 500 µmol/L H_2_O_2_ for 3 h. The optimal concentrations for E and M (200 μg/mL) were selected, as was the lowest treatment concentration that improved cell viability of ARPE-19 cells exposed to H_2_O_2_. Based on these findings, we used a mix composed of 40 μg/mL Saffron, 200 μg/mL Elderberry, and 200 μg/mL Melilotus for the subsequent experiments. In the further experiments, the growth medium was replaced after 3 days with medium containing Affroneye alone (SAF) or a mix of Affroneye, Eldosamb and *Melilotus officinalis* (SEM mix). The next day, 500 µmol/L H_2_O_2_ was added to the culture medium for further 3 h, then cells were processed for each analysis (Figure 9). The morphological characteristics of ARPE-19 cells under each experimental condition documented using light microscopy (Olympus CK 2 Inverted Phase Contrast Microscope, Olympus Italia S.r.l., Milan, Italy).

### 4.2. Cell Viability Assays

In order to evaluate cell proliferation, ARPE-19 cells were seeded on 96-well plates (10 × 10^3^ cells/well) and cultured as described. Viable cells were determined using the Cell Titer 96 Aqueous One Solution Cell Proliferation Assay (Promega, Madison, WI, USA) according to the manufacturer’s instructions. Absorbance was recorded using a microplate reader (TecanTecan Trading AG, Männedorf, Switzerland) at the wavelength of 490 nm. Results were expressed as percentage of absorbance in each treatment compared to absorbance value of the control (CTR, 100%).$$Cell\;viability\;\left(\%\right)=\frac{{OD}_{sample}-{OD}_{blank}}{{OD}_{control}-{OD}_{blank}}\times 100\%$$

### 4.3. Caspase-3 Activity

The activity of caspase-3 was measured using the Caspase-Glo^®^ 3/7 Assay (Promega, Madison, WI, USA), according to the procedure described in the kit. In brief, cells were seeded into 96-well culture plates at 10 × 10^3^ cells/well and treated as previously described, then a luminogenic caspase-3/7 substrate, which contains the tetrapeptide sequence DEVD, was added to each well in a reagent optimized for caspase activity, luciferase activity and cell lysis. The luminescent signal produced by luciferase was recorded using the plate-reading luminometer Glomax (Promega, Madison, WI, USA). Results were expressed as percentage of luminescence compared to control (CTR, 100%).

### 4.4. LDH Release

LDH release from non-viable cells was quantified using the LDH-Glo™ Cytotoxicity Assay (Promega, Madison, WI, USA), adhering to the manufacturer’s protocol. In brief, cells were seeded into 96-well culture plates at 10 × 10^3^ cells/well and treated as previously described. Then, a small amount of culture medium (2–5 µL) was removed and diluted into LDH Storage Buffer. LDH activity was measured by adding an equal volume of LDH Detection Reagent to the diluted sample. The luminescent signal generated was proportional to the amount of LDH in the sample. Results were expressed as percentage of luminescence compared to control (CTR, 100%).

### 4.5. Evaluation of Intracellular Reactive Oxygen Species (ROS)

The levels of ROS was assessed using a Reactive Oxygen Species Assay Kit (Promega, Madison, WI, USA), adhering to the manufacturer’s protocol. In brief, cells were seeded into 96-well culture plates at 10 × 10^3^ cells/well and treated as previously described, and then substrate solution was added to each well. Subsequently, the media samples were transferred to a separate plate and combined with an equal volume of Detection solution. The relative luminescence units were recorded using a plate-reading luminometer (Glomax, Promega, Madison, WI, USA). Results were expressed as percentage of luminescence compared to control (CTR, 100%).

### 4.6. Measuring of Glutathione (GSH) Levels

Detection of GSH in ARPE-19 cells plated in 96-well dishes (10 × 10^3^ cells per well) and cultured as described above was performed using the GSH/GSSG-Glo™ Assay (Promega, Madison, WI, USA) according to the manufacturer’s instructions. Briefly, the assay is a luminescence-based system to detect and quantify total glutathione (GSH + GSSG), GSSG and GSH/GSSG ratios. The assay is based on the conversion catalyzed by glutathione S-transferase (GST) of a luciferin derivative into luciferin in the presence of glutathione. The relative luminescence units were recorded using a plate-reading luminometer (Glomax, Promega, Madison, WI, USA). GSH/GSSG ratios were calculated directly from luminescence measurements.

### 4.7. Immunoblotting Analysis

At the end of the experiments a batch of ARPE-19 cells were lysed in RIPA buffer supplemented with protease and phosphatase inhibitors (Pierce, Rockford, MD, USA) and protein concentrations were determined using the BCA Protein Assay Kit. Thirty micrograms of total cell proteins were separated on 4–20% SDS-PAGE gradient gels (Life Technologies Italia, Milan, Italy) and transferred onto nitrocellulose using iBlot system 3 (Life Technologies Italia, Milan, Italy). Filters were blocked in Protein Free T20 Blocking Buffer (Pierce Biotechnology, Rockford, IL, USA) and incubated overnight at 4 °C with primary antibodies specific to NRF2 (D1Z9C), Keap-1 (D6B12), SOD2 (D3X8F) and β-ACT (8H10D10 cat. 3700) from Cell Signaling Technology, Beverly, MA, USA. Secondary specific horseradish-peroxidase linked antibodies (Cell Signaling Technology, Beverly, MA, USA) were added and left for 1 h at room temperature. Bound antibodies were detected using the enhanced chemiluminescence lighting system (LiteAblot EXTEND, Euroclone, Milan, Italy), according to the manufacturer’s instructions. Bands of interest were quantified by densitometry using Alliance LD2 with Uvitec 1D software (v18.07, UVITEC Ltd., Cambridge, UK). The results were expressed as a percentage of CTR (defined as 100%).

### 4.8. Caspase-1 Activity

Caspase-1 activity was evaluated using the Caspase-Glo^®^ 1 Inflammasome Assay (Promega, Madison, WI, USA), according to the manufacture’s instruction. In brief, ARPE-19 cells plated in 96-well dishes (10 × 10^3^ cells per well) and cultured as described above. Then, the luminogenic caspase-1 substrate, Z-WEHD-amino-luciferin, was diluted in a lytic reagent and added to each well. This results in cell lysis, substrate cleavage by caspase-1 and generation of light by a recombinant luciferase, which is proportional to caspase-1 activity. The relative luminescence units were recorded using a plate-reading luminometer (Glomax, Promega, Madison, WI, USA). Results were expressed as percentage of luminescence compared to control (CTR, 100%).

### 4.9. Interleukin-1β Secretion

To evaluate IL-1β secretion ARPE-19 cells were cultured for 24 h in serum-free medium containing SAF or SEM mix before treatment for 3 h with 500 μmol/L H_2_O_2_. Then, the conditioned media were collected and stored at −80 °C until the assay was performed. Quantitative secretion of IL-1β was evaluated by ELISA (RayBiotech Life, Inc., Peachtree Corners, GA, USA). Concentration of IL-1β was extrapolated from the standard curve and each value was related to total protein concentration of the respective lysate. Protein content of lysates was determined using the BCA Protein Assay Kit (Pierce, Rockford, MD, USA) according to the manufacturer’s instructions.

### 4.10. Statistical Analysis

The results are representative of at least 3 experiments. All analyses were carried out with GraphPad Prism 9.0.0 software (GraphPad Software, San Diego, CA, USA). Quantifications were expressed as the mean ± SD. Data of more groups were expressed as the mean ± SD and then analyzed using one-way ANOVA followed by Dunnett’s or Tukey’s multiple comparison test. *p* value < 0.05 was considered as statistically significant.

## Figures and Tables

**Figure 1 ijms-26-01496-f001:**
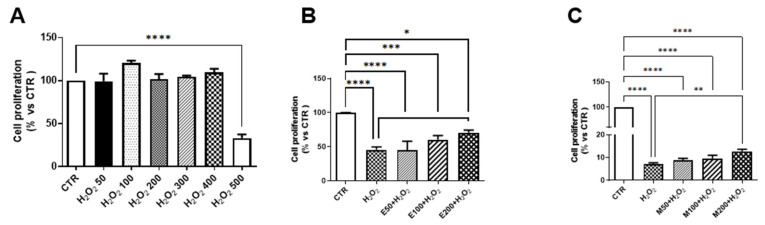
Arpe-19 cell viability. (**A**) Cell viability of ARPE-19 cells exposed for 3 h to 50, 100, 200, 300, 400 and 500 µmo/L of H_2_O_2_ (n = 8); (**B**,**C**) Viability of ARPE-19 cells cultured for 24 h respectively with 50, 100 or 200 µg/mL of *Elderberry* (E) and *Melilotus* (M) and then exposed for 3 h to 500 µmol/L H_2_O_2_ (n = 4). The results are expressed as percentage versus CTR. * *p* < 0.05, ** *p* < 0.01, *** *p* < 0.001 and **** *p* < 0.0001.

**Figure 2 ijms-26-01496-f002:**
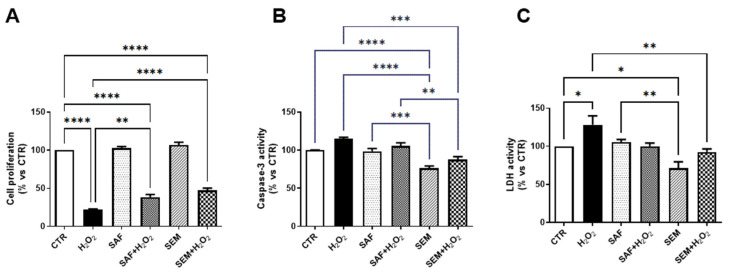
Cell viability of ARPE-19 cells cultured for 24 h with SAF and SEM mix and then exposed for 3 h to 500 µmol/L H_2_O_2_. (**A**) Cell proliferation rate. Evaluation of activity of (**B**) caspase-3 and of (**C**) LDH. The results are expressed as percentage versus CTR (n = 8). * *p* < 0.05, ** *p* < 0.01, *** *p* < 0.001 and **** *p* < 0.0001.

**Figure 3 ijms-26-01496-f003:**
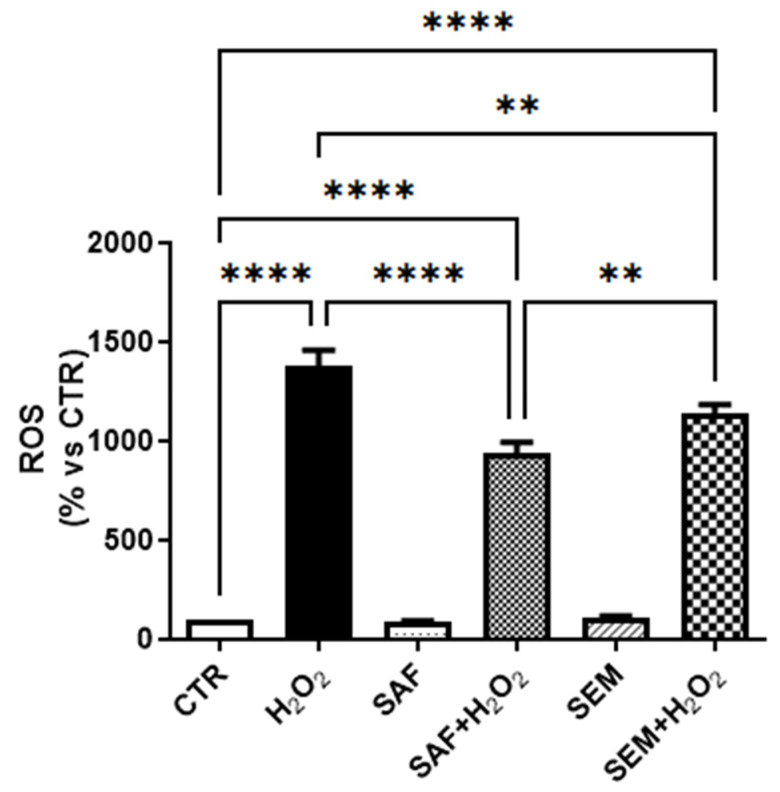
ROS production in ARPE-19 cells cultured for 24 h with SAF and SEM mix and then exposed for 3 h to 500 µmol/L H_2_O_2_. The results are expressed as percentage of luminescence compared to control (CTR, 100%) (n = 6) ** *p* < 0.01 and **** *p* < 0.0001.

**Figure 4 ijms-26-01496-f004:**
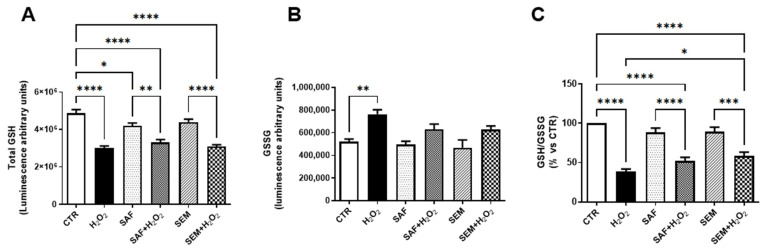
Amount of total glutathione (**A**), oxidized GSSG (**B**) and GSSG/GSH ratio (**C**) in ARPE-19 cells cultured for 24 h with SAF and SEM mix and then exposed for 3 h to 500 µmol/L H_2_O_2_. The results are expressed as percentage of luminescence compared to control (n = 8). * *p* < 0.05, ** *p* < 0.01, *** *p* < 0.001 and **** *p* < 0.0001.

**Figure 5 ijms-26-01496-f005:**
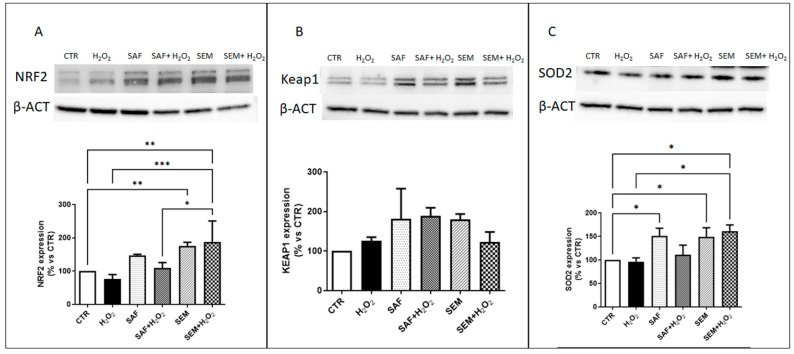
Representative Western blot analysis of protein expression of (**A**) NRF2, (**B**) Keap1 and (**C**) SOD2 in ARPE-19 cells cultured for 24 h with SAF and SEM mix and then incubated for 3 h with 500 µmol/L H_2_O_2_. In the upper part of the panel are representative images of western blots, and in the lower part of the panel quantification of densitometries of relative bands. The results are expressed as percentage compared to control (CTR, 100%) (n = 6). * *p* < 0.05, ** *p* < 0.01 and *** *p* < 0.001.

**Figure 6 ijms-26-01496-f006:**
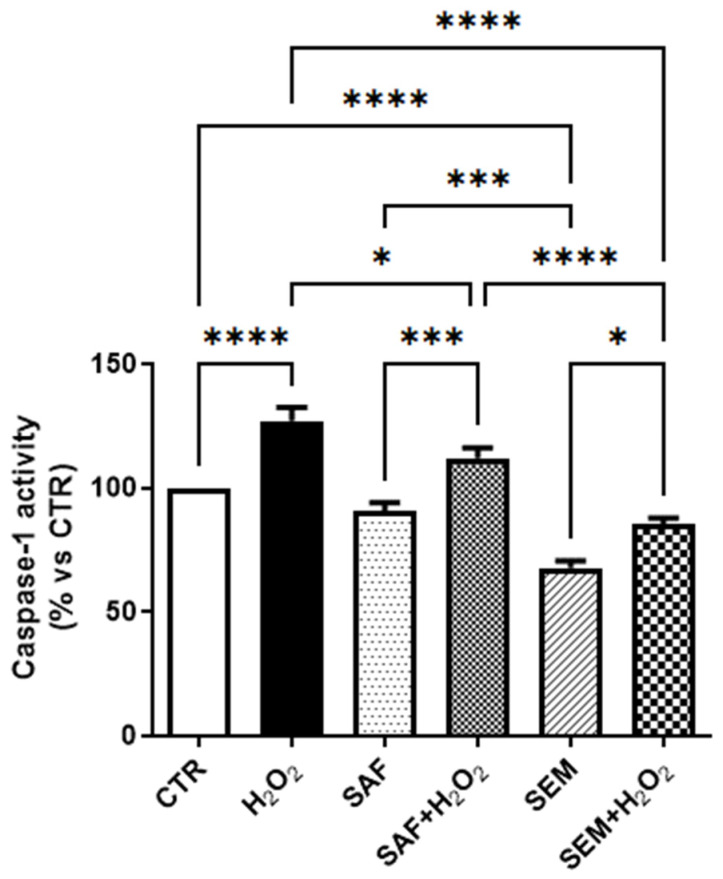
Caspase-1 activity in ARPE-19 cells cultured for 24 h with SAF and SEM mix and then incubated for 3 h with 500 µmol/L H_2_O_2_. The results are expressed as percentage of luminescence compared to control (CTR, 100%) (n = 6). * *p* < 0.05, *** *p* < 0.001 and **** *p* < 0.0001.

**Figure 7 ijms-26-01496-f007:**
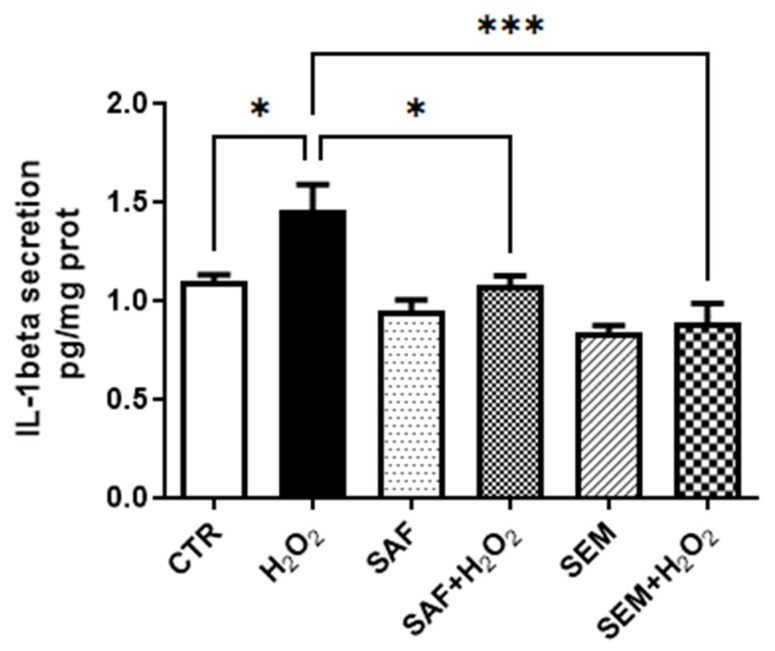
Secretion of IL-1β in ARPE-19 cells cultured for 24h with S and SEM mix and then incubated for 3 h with 500 µmol/L H_2_O_2_. IL-1β content in the collected media was assayed by ELISA. Data were normalized to cellular protein content. Each bar represents the mean ± SD (n = 4). * *p* < 0.05 and *** *p* < 0.001.

**Figure 8 ijms-26-01496-f008:**
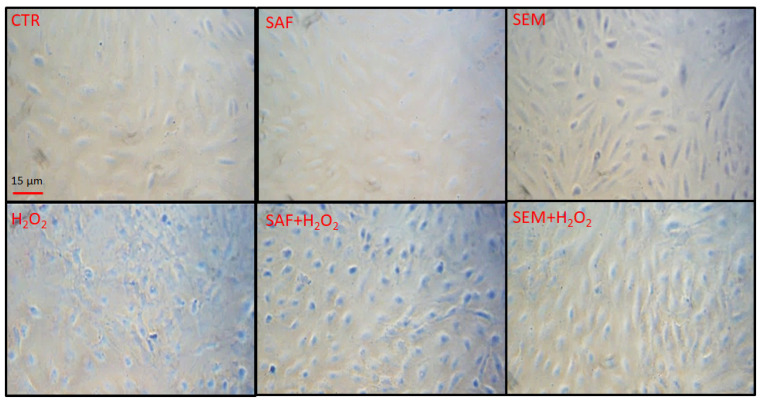
Representative morphological images of ARPE-19 cells: the upper panel represents ARPE-19 cells cultured for 24 h in standard medium (CTR), and in media containing SAF or SEM mix; the lower panel represents the same batches of cells after exposure to 500 µmol/L H_2_O_2_ for 3 h. Red scale bar corresponds to 15 μm.

**Figure 9 ijms-26-01496-f009:**
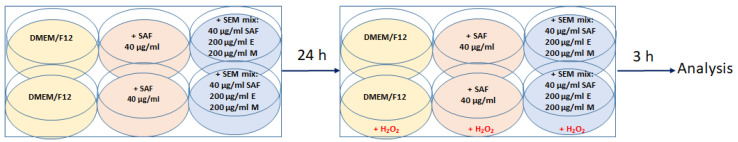
Schematic representation of cell treatments.

## Data Availability

All the data are contained within the article.

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
