# Peer review of "Combination of *Saffron* (*Crocus sativus*), *Elderberry* (*Sambucus nigra* L.) and *Melilotus officinalis* Protects ARPE-19 Cells from Oxidative Stress"

_ijms, 2025, doi:10.3390/ijms26041496_

Round 1

Reviewer 1 Report

Comments and Suggestions for Authors

Summary

The authors present an in-depth evaluation of ROS production in ARPE-19 cells treated with saffron, elderberry, and melilotus, either individually or in combination. While the study is intriguing and provides meaningful data, the discussion requires further elaboration to highlight the novelty and implications of the findings. The introduction should briefly include additional information about the plant species under investigation. Furthermore, details about the source and preparation of the treatments must be clarified, and the results section would benefit from more comprehensive explanations.

Title

Consider revising the title to include specific details about the treatments used in the study. This will give readers a clearer understanding of the work's focus.

Abstract

  • In the final lines, underscore the novelty of the study and provide a "take-home" message for readers.
  • Clearly articulate the potential implications of your findings in a concise manner.

Introduction

  • Briefly present the three plant species, including their Latin names, to provide context on their pharmacological relevance. This will aid readers in identifying the key secondary metabolites responsible for their beneficial properties.
  • Consider incorporating the following references for a broader perspective on the species discussed:

-        https://www.actahort.org/books/1354/1354_8.htm

-        10.1021/acs.jafc.2c00010

-        https://doi.org/10.1002/ptr.875

Results

  • Lines 105-106: Move this section to the Discussion for better alignment with the overall narrative.
  • Lines 106-107: Expand the description to provide a more detailed account of the results.
  • Lines 114-118: Shift this paragraph to the Discussion, leaving the Results section focused on trends and variations observed in the experiments.
  • Replace the use of the dollar sign ($) (despite it is funny) to denote significance with asterisks (*) for a more professional tone.
  • Please, You should enhance the graphical quality of the bar plots. Ensure figure captions are detailed and briefly summarize the observed trends in the experiments.

Discussion

  • Include and discuss the activity of saffron against ROS and reactive nitrogen species (RNS), referencing prior studies such as:

-        https://doi.org/10.3390/antiox8070224

-        10.4103/1673-5374.274325

  • Discuss the roles of ROS-regulating enzymes like superoxide dismutase (SOD), catalase (CAT), and ascorbate peroxidase (APX) in the context of your findings.
  • Lines 209-210: Revise this paragraph to present a more compelling narrative about the broader impact of your study. For instance, emphasize how your findings could inspire the development of new therapeutic strategies for ocular pathologies, rather than limiting the discussion to dietary supplementation.

Materials and Methods

  • Clarify the preparation of treatments:

-        Were the saffron, elderberry, and melilotus formulations commercial products? If so, specify the provider and the exact product details.

-        If raw plant material was used, describe the extraction and preparation methods in detail (e.g., dried leaves, saffron stigmas).

-        If VISUfarma spa, Roma, Italy supplied the treatments, provide more information on the specific formulations and typologies.

  • For subsection 4.8 (ANOVA analysis):

-        Specify the type of ANOVA performed (e.g., one-way, two-way).

-        Clearly indicate the experimental factors and levels tested.

Author Response

We thank you very much for taking the time to review this manuscript. Please find the detailed responses below and the corresponding revisions/corrections highlighted/in track changes in the re-submitted files.

The authors present an in-depth evaluation of ROS production in ARPE-19 cells treated with saffron, elderberry, and melilotus, either individually or in combination. While the study is intriguing and provides meaningful data, the discussion requires further elaboration to highlight the novelty and implications of the findings. The introduction should briefly include additional information about the plant species under investigation. Furthermore, details about the source and preparation of the treatments must be clarified, and the results section would benefit from more comprehensive explanations.

Title

Comment 1: Consider revising the title to include specific details about the treatments used in the study. This will give readers a clearer understanding of the work's focus.

Response 1: we changed the title as follow: “Combination of Saffron (Crocus Sativus), Elderberry (Sambucus nigra L.) and Melilotus Officinalis protects ARPE-19 cells from oxidative stress”

Abstract

  • I Comment 2: In the final lines, underscore the novelty of the study and provide a "take-home" message for readers.
  • Clearly articulate the potential implications of your findings in a concise manner.

Response 2: we enhanced the abstract by summarizing the new results and added the following sentence at the end of the abstract:  “These findings suggest that the combination of Saffron, Elderberry and Melilotus could have therapeutic potential in the prevention and treatment of retinal degenerative diseases.”

Introduction

  • Comment 3: Briefly present the three plant species, including their Latin names, to provide context on their pharmacological relevance. This will aid readers in identifying the key secondary metabolites responsible for their beneficial properties.

Response 3: We briefly presented the three plants species in the second part of the Introduction: “Saffron (Crocus Sativus) is one of the most characterized natural compound with an-ti-inflammatory proprieties [22]. It is an annual species mainly produced in Iran and in Mediterranean countries traditionally used as a food additive and aromatic spice [23]. The therapeutic activities of Saffron are due to its main bioactive components crocetin, a carot-enoid compound, and crocin that derives from crocetin hydrolyzation during the intesti-nal absorption [24, 25]…..” ……. “…Elderberry is a species complex of flowering plants in the family Adoxaceae with many health functions [30]. Both flowers and fruits contain several bioactive compounds, including phenolic acids, flavanones, flavonols, and anthocyanins, responsible of the an-ti-inflammatory and antioxidant properties [31-34]. Melilotus Officinalis belongs to family Fabaceae and is used as herbal medicine from ancient times [35, 36]. Their beneficial effects are mainly due to the biological activity of the coumarins formed during the drying process [37, 38]. Despite the encouraging results in literature, the mechanisms of action by which these natural compounds provide beneficial effects in RPE cells are not yet completely understood.

  •  
  • Comment 4: Consider incorporating the following references for a broader perspective on the species discussed:

-        https://www.actahort.org/books/1354/1354_8.htm

-        10.1021/acs.jafc.2c00010

-        https://doi.org/10.1002/ptr.875

Response 4: we included these references (numbered 23, 35, 36) in the presentation of the three plants species (See response 3).

Results

  • Comment 5: Lines 105-106: Move this section to the Discussion for better alignment with the overall narrative.

Response 5: we move lines 105-160 to the first part of the discussion section.: “However, when the balance between oxidative and reduction processes cannot be maintained by endogenous antioxidant defences, it is necessary to help cells with therapeutic antioxidant strategies. An imbalance between generation of reactive ROS and antioxidant defense systems represents the primary cause of cell dysfunction [54]. …

  •  
  • Comment 6: Lines 106-107: Expand the description to provide a more detailed account of the results.

Response 6: As suggested, we expanded the description with more details in section 2.2. Saffron and SEM mix reduced oxidative stress: “To evaluate whether cytoprotective effects of SAF and SEM mix on ARPE-19 cells exposed to 500 µmol/L H2O2 were related to their antioxidant proprieties, we evaluated oxidative stress by measuring the levels of ROS in cell culture medium. Exposure of ARPE-19 cells to H2O2 induced a significant increment in ROS production in comparison to cells incubated with control medium (Figure 3). ….

  •  
  • Comment 7: Lines 114-118: Shift this paragraph to the Discussion, leaving the Results section focused on trends and variations observed in the experiments.

Response 7: We appreciate your feedback and agree with your comment. However, we have decided to retain this paragraph to allow for a better interpretation of the results.

  • Comment 8: Replace the use of the dollar sign ($) (despite it is funny) to denote significance with asterisks (*) for a more professional tone.

Response 8: we replaced the dollar sign with asterisks in all the figures.

  •  
  • Comment 9: Please, You should enhance the graphical quality of the bar plots. Ensure figure captions are detailed and briefly summarize the observed trends in the experiments.

Response 9: We improved the graphical quality of the plot by using black-and-white styles and revised the figure captions to include the missing information.

Discussion

  • Comment 10: Include and discuss the activity of saffron against ROS and reactive nitrogen species (RNS), referencing prior studies such as:

-        https://doi.org/10.3390/antiox8070224

-        10.4103/1673-5374.274325

Response 10: We included and discussed the activity of Saffron against ROS and RNS in the third paragraph of the discussion: “Therefore, treatment with antioxidant may be designed to neutralize free radicals and in-crease the antioxidant defence system. The antioxidant proprieties of Saffron arise from both its direct action as a free radical scavenger, and its regulation of genes involved in the antioxidant response [22, 25, 55]. Indeed, Saffron has been shown to activate multiple pathways, including the inhibition of free radical production, reduction of lipid peroxidation and an increase in the levels and activity of antioxidant enzymes. Recently Di Marco et al. demonstrated that Saffron administration reduced retinal degeneration in patients with AMD, suggesting that Saffron may have potential added benefits compared to traditional antioxidant treatments [56]. In line with these findings, we found that Saffron alone or in combination with Elderberry and Melilotus was able to reduce ROS production.…

  • Comment 11: Discuss the roles of ROS-regulating enzymes like superoxide dismutase (SOD), catalase (CAT), and ascorbate peroxidase (APX) in the context of your findings.

Response 11: we discussed the role of these enzymes in the third paragraph of the discussion: “It is well known that RPE cells, due to their high rate of oxygen (O2) consumption, developed an efficient antioxidant machinery to counteract ROS production during metabolic processes [57, 58]. Among the key antioxidant enzymes, Superoxide dismutase (SOD) and Catalase (CAT) play essential roles in protecting against superoxide radicals: SOD neutralizes superoxide anion converting it to hydrogen peroxide (H2O2) and O2, then CAT completes the detoxification by reducing H2O2 into water and oxygen [59, 60]..”

  •  
  • Comment 12: Lines 209-210: Revise this paragraph to present a more compelling narrative about the broader impact of your study. For instance, emphasize how your findings could inspire the development of new therapeutic strategies for ocular pathologies, rather than limiting the discussion to dietary supplementation.

Response 12: we wrote the conclusions in a separate paragraph and emphatized the development of new therapeutic strategies involving the use of these natural compounds: “In conclusion, our results demonstrate that both Saffron alone and its combination with Elderberry and Melilotus are able to counteract the damaging effects of hydrogen peroxide. However, the mix is more effective in preventing inflammation, suggesting that the combination of different bioactive compounds may have stronger antioxidant protection. These findings provide evidence that adding saffron, Elderberry and Melilotus to the culture medium confers cytoprotective effects on RPE cells. In particular, this combination activates both antioxidant and anti-inflammatory pathways, restoring the redox balance and preserving cell morphology. Overall, our results suggest that drug formulations containing combination of Saffron, Elderberry and Melilotus, whether developed for topical use or as dietary supplement, may offer potential benefits in the prevention and treatment of retinal degenerative diseases.

  •  

Materials and Methods

  • Comment 13: Clarify the preparation of treatments:

-        Were the saffron, elderberry, and melilotus formulations commercial products? If so, specify the provider and the exact product details.

-        If raw plant material was used, describe the extraction and preparation methods in detail (e.g., dried leaves, saffron stigmas).

-        If VISUfarma spa, Roma, Italy supplied the treatments, provide more information on the specific formulations and typologies.

Response 13: we clarified the preparation of treatments and added the information required and a representative figure (number 9) in section 4.1. Cell culture and experimental conditions: ” Afterwards, cells were pretreated for 24 hours with 40 μg/ml of Saffron (Crocus sativus L.) standardized to contain 3% crocins (Affroneye, Pharmactive Biotech Products, Alco-bendas, Madrid, Spain) (SAF) or various doses (0, 50, 100 and 200 µg/ml) of the high-quality elderberry plant extract Eldosamb® (Sambucus nigra L., Anklam Extrakt GmbH, Anklam, Germany) (E) or of Melilotus Officinalis containing 12% cumarin (Nutraceutica srl, Monterenzio, Bologna, Italy) (M) (all the compounds were kindly gifted by VISUfarma spa, Roma, Italy). Then ARPE-19 cells were exposed to 500 µmol/L H2O2 for 3 hours. The optimal concentrations for E and M (200 μg/ml) were selected, as was the lowest treatment concentration that improved cell viability of ARPE-19 cells exposed to H2O2. In the next experiments, the growth medium was replaced after 3 days with medium containing Affroneye (S) or a mix of Affroneye, Eldosamb and Melilotus Officinalis (SEM mix). The next day, 500 µmol/L H2O2 was added to the culture medium for further 3 hours, then cells were processed for each analysis.”

  • Comment 14: For subsection 4.8 (ANOVA analysis):

-        Specify the type of ANOVA performed (e.g., one-way, two-way).

Response 14: we reported that performed one-way ANOVA in section 5.10 Statistical Analysis.

-       Comment 15:  Clearly indicate the experimental factors and levels tested.

Response 15: we clarified experimental factors and levels tested information in in section 5.1 Cell culture and experimental conditions and in figure 9.

Reviewer 2 Report

Comments and Suggestions for Authors

The manuscript "Combination of Saffron With Elderberry and Melilotus Blunted Oxidative Damage in ARPE-19 Cells" is interesting as it explores the potential beneficial effect of a mixture against hydrogen peroxide-induced damage in ARPE-19 cells. The concept of the work is certainly relevant to the field and the results indicate the possible use of this formulation in the future. While the findings appear promising, there are some key suggestions for improving the work.

# The abstract is too general and should include more information regarding the methods used and the results.

# The concentration of hydrogen peroxide used to induce damage to the ARPE-19 cell line should be revised. While the concentration of 200 umol did not alter cell viability, 500 umol decreased viability to less than 50% (Figure 1A), which seems challenging to recover from. Authors should calculate the IC50 for hydrogen peroxide and include intermediate concentrations.

# Currently, there is insufficient evidence to understand the molecular mechanisms of the isolated components or their mixture against the damage caused by hydrogen peroxide to ARPE-19 cells. Authors should conduct additional experiments, including more analysis, especially through Western blot and qPCR protocols.

# The authors claim to have examined the inflammatory status, but only Caspase-1 and 3 activities were assessed and could be more directly related to this end. This is insufficient to provide a picture of the possible anti-inflammatory action and the molecular mechanisms of the mixture. The authors state in the conclusion of the work that the mixture prevents inflammation, however, this is not supported by their current findings. Additionally, why are the results of Caspase-1 and 3 separated in different images?

# The description of the methods is lacking vital information. For example, there is not enough information regarding the obtaining of saffron, Elderberry, and Melilotus. Were these extracts, powders? Were the concentrations and exposure periods used based on the literature? Is there information on the concentrations of compounds in these samples? Provide clarification.

# The graphs use a mix of colors that make the figures unattractive. The figure legends do not provide enough information to fully comprehend the images. The meaning of acronyms is not provided, nor are the concentrations and units of hydrogen peroxide and the mixture components. Thus, the reader must go back and forth in the text to understand the figures.

# In Figure 6 the scale bars are missing and it is also difficult to distinguish differences between the groups. Please improve the quality of these images.

# Write scientific names always in italics.

Author Response

We thank you very much for taking the time to review this manuscript. Please find the detailed responses below and the corresponding revisions/corrections highlighted/in track changes in the re-submitted files.

The manuscript "Combination of Saffron With Elderberry and Melilotus Blunted Oxidative Damage in ARPE-19 Cells" is interesting as it explores the potential beneficial effect of a mixture against hydrogen peroxide-induced damage in ARPE-19 cells. The concept of the work is certainly relevant to the field and the results indicate the possible use of this formulation in the future. While the findings appear promising, there are some key suggestions for improving the work.

Comment 1: The abstract is too general and should include more information regarding the methods used and the results.

Response 1: due to the abstract's word limit (200 words), we enhanced it by summarizing the new results, but were unable to include additional details about the methods.

Comment 2:  The concentration of hydrogen peroxide used to induce damage to the ARPE-19 cell line should be revised. While the concentration of 200 umol did not alter cell viability, 500 umol decreased viability to less than 50% (Figure 1A), which seems challenging to recover from. Authors should calculate the IC50 for hydrogen peroxide and include intermediate concentrations.

Response 2: Hydrogen peroxide (H2O2) is commonly used to assess the protective effects of antioxidants against oxidative stress in RPE cells. The concentration of 500 µmol/L may induce mild to moderate oxidative stress, resulting in several cellular responses, which space from inflammation to cell death, depending on the exposure time. To verify whether 500 µmol/L H2O2 is a suitable concentration in our experimental setting, we performed cell viability tests including other intermediate concentrations of H2O2 (300 and 400 µmol). The results showed that only the greater concentation of H2O2 (500 µmol/L) significantly reduced proliferation of ARPE-19 cells, confirming our previous results. We included these data in Figure 1. In addition, we verified whether cells exposed to 500 µmol/L H2O2 are able to recovery. To this aim, we exposed ARPE-19 cells for 3 hours to 500 µM H2O2; then we replaced the medium, cultured the cells for further 24 hours and tested cell viability. We found that cells were almost completely recovered (about 77%). Therefore, despite 500 µmol/L of H2O2 strongly reduced cell viability, ARPE-19 cells are able to recovery from this oxidative stress within 24 hours. We included these data in supplementary figure 1.

Comment 3: Currently, there is insufficient evidence to understand the molecular mechanisms of the isolated components or their mixture against the damage caused by hydrogen peroxide to ARPE-19 cells. Authors should conduct additional experiments, including more analysis, especially through Western blot and qPCR protocols.

Response 3: to investigate the mechanisms potentially involved in the antioxidant response of ARPE-19 cells we explored the activation of the Nuclear Factor Erythroid 2-Related Factor 2 (NRF2) pathway, a key transcription factor that regulates the expression of various antioxidant enzymes [46]. The transcriptional activity of NRF2 plays a key role in helping RPE cells to counteract oxidative stress. The ability of NRF2 to translocate into the nucleus depends on its interaction with Keap1. In response to oxidative stress, NRF2 dissociates from Keap1 and translocates into the nucleus, where it binds to antioxidant response NRF2elements (AREs) in the promoter region of several antioxidant genes, including Superoxide Dismutase 2 (SOD2). To explore the molecular mechanisms underlying the protective effects of SAF and SEM mix, we investigated whether these treatments modulate the protein expression of these factors. The results are showed in figure 5.

Comment 4:  The authors claim to have examined the inflammatory status, but only Caspase-1 and 3 activities were assessed and could be more directly related to this end. This is insufficient to provide a picture of the possible anti-inflammatory action and the molecular mechanisms of the mixture. The authors state in the conclusion of the work that the mixture prevents inflammation, however, this is not supported by their current findings. Additionally, why are the results of Caspase-1 and 3 separated in different images?

Response 4: We evaluated the activity of caspase-1 to determine whether the SAF and SEM mix could activate anti-inflammatory pathways in ARPE-19 cells exposed to oxidative stress. Upon activation, caspase-1 plays a crucial role in inflammation by cleaving the precursor forms of several cytokines into their active forms, which are then released by the cells. Interleukin-1β (IL-1β), one of the key substrates of caspase-1, is considered a major pro-inflammatory mediator. To further support the anti-inflammatory properties of the SAF and SEM mix, we measured IL-1β levels in the culture medium of ARPE-19 cells. The results confirm that the reduction in caspase-1 activity was associated with decreased IL-1β secretion in cells pretreated with the SEM mix. The results are shown in figures 6 and 7. Finally, caspase-1 is primarily involved in inflammation and the activation of inflammatory cytokines, so we assessed its activity as a marker of inflammation. In contrast, caspase-3 directly triggers the breakdown of cellular components, leading to controlled, non-inflammatory cell death, and we evaluated its activation as a marker of apoptosis. Therefore, we presented their activation in different sections of the results.

Comment 5:  The description of the methods is lacking vital information. For example, there is not enough information regarding the obtaining of saffron, Elderberry, and Melilotus. Were these extracts, powders? Were the concentrations and exposure periods used based on the literature? Is there information on the concentrations of compounds in these samples? Provide clarification.

Response 5: we added these information in the section section 5.1. Cell culture and experimental conditions:” Afterwards, cells were pretreated for 24 hours with 40 μg/ml of Saffron (Crocus sativus L.) standardized to contain 3% crocins (Affroneye, Pharmactive Biotech Products, Alcoben-das, Madrid, Spain) (SAF) or various doses (0, 50, 100 and 200 µg/ml) of the high-quality elderberry plant extract Eldosamb® (Sambucus nigra L., Anklam Extrakt GmbH, Anklam, Germany) (E) or of Melilotus Officinalis containing 12% cumarin (Nutraceutica srl, Mon-terenzio, Bologna, Italy) (M) (all the compounds were kindly gifted by VISUfarma spa, Roma, Italy). Then ARPE-19 cells were exposed to 500 µmol/L H2O2 for 3 hours. The opti-mal concentrations for E and M (200 μg/ml) were selected, as was the lowest treatment concentration that improved cell viability of ARPE-19 cells exposed to H2O2. In the next experiments, the growth medium was replaced after 3 days with medium containing Affroneye (S) or a mix of Affroneye, Eldosamb and Melilotus Officinalis (SEM mix). The next day, 500 µmol/L H2O2 was added to the culture medium for further 3 hours, then cells were processed for each analysis.”

Given the lack of information on the effects of these natural compounds on RPE cells, we conducted experiments replicating the conditions described by Di Paolo (PMID 36838685 : Cells pretreated for 24 h with the various saffron samples at 40 μg/ml and then exposed for 3 h to H2O2 500 μmol/L). This allowed us to compare our results with his previous findings. Next we tested the viability of cells exposed for 3 h to 0, 50, 100, 200, 300, 400 and 500 µM H2O2. Once confirmed the results of Di Paolo et al., we investigated the protective effects of Elderberry and Melilotus on viability of ARPE-19 cells testing concentrations ranging from 50 to 200 µg/ml. The results showed that only the highest concentration (200 μg/ml) of both compounds significantly improved cell proliferation when cells are exposed to 500 μmol/L of H2O2. Based on these findings, we used a mix composed of 40 μg/ml Saffron, 200 μg/ml Elderberry, and 200 μg/ml Melilotus for the subsequent experiments.

Comment 6:  The graphs use a mix of colors that make the figures unattractive. The figure legends do not provide enough information to fully comprehend the images. The meaning of acronyms is not provided, nor are the concentrations and units of hydrogen peroxide and the mixture components. Thus, the reader must go back and forth in the text to understand the figures.

Response 6: we redrew the graphs using black and white styles and completed the figure legends with the missing informations.

Comment 7:  In Figure 6 the scale bars are missing and it is also difficult to distinguish differences between the groups. Please improve the quality of these images.

Response 7: we improved the quality of the images and added the scale bars.

Comment 8:  Write scientific names always in italics.

Response 8: we checked the text and wrote all scientific names in Italic

Reviewer 3 Report

Comments and Suggestions for Authors

Dear Editor,

I have reviewed the manuscript titled ‘’Combination of Saffron With Elderberry and Melilotus Blunted Oxidative Damage in ARPE-19 Cells’ The manuscript can be resubmitted after revision.

-Title ‘’Combination of Saffron With Elderberry and Melilotus Blunted Oxidative Damage in ARPE-19 Cells’’ blunted is the not right scientific verb, replace it with suitable other verb.

-Introduction:Lines 29-33: The resulting secretion of early hallmarks of inflammation, such as MCP-1, interleukin (IL)-1β and IL-18, activates several intracellular signals, which, in turn, stimulate the production of other cytokines, leading to recruitment of inflammatory cells, thus generating a vicious cycle that rises inflammation and exposes retinal cells to oxidative damages. Running on sentence. Rewrite with proper scientific verbs and remove terminology such as ‘recruitment’. Add the references.

-In introduction, add one paragraph about free radicals responsible for DR and AMD and their mechanism of action.

-Saffron With Elderberry and Melilotus- what form these components were used? Extract or whole plants powder mixture extract or separately took then mixed together.

-How did the authors decide the concentration of saffron will be effective at 40 μM?

Why did authors not use any standard compound that used to treat oxidative stress to compare the results?

-4.2. Cell viability assays: elaborate the proper procedure with concentration of treatment used for cell viability analysis. Write the formula of cell viability calculations

-Figure 1C. Arpe-19 cell viability redraw and make uniform ARPE-19 representation. Use the same y-axis intervals as for figure 1a and b used.

-Figure 6. Representative morphological images of ARPE-19 cells cultured for 24h with S and SEM (upper panel) and then exposed to OX (lower panel). Why did not upper panel cell culture expose together with OX? Does disease occur first or treatment giving before disease occurrence?

- Write the conclusion separately.

- 36% plagiarism was found, so reduce the plagiarism to 15%.

Below given sections are plagiarized. Introduction is partial plagiarized while material methods section is mostly plagiarized. Also figure legends were showing plagiarism.

1. Introduction

4. Materials and Methods

4.1. Cell culture and experimental conditions

4.2. Cell viability assays

4.3. Caspase-3 activity

4.4. LDH release

4.5. Evaluation of intracellular Reactive Oxygen Species (ROS)

4.6. Measuring of glutathione (GSH) levels

4.7. Caspase-1 activity

4.8. Statistical analysis 

Comments on the Quality of English Language

- Check the manuscript for non-scientific terminology, grammatical errors, and sentence errors.

Author Response

We thank you very much for taking the time to review this manuscript. Please find the detailed responses below and the corresponding revisions/corrections highlighted/in track changes in the re-submitted files.

Dear Editor,

I have reviewed the manuscript titled ‘’Combination of Saffron With Elderberry and Melilotus Blunted Oxidative Damage in ARPE-19 Cells’ The manuscript can be resubmitted after revision.

Comment 1: Title ‘’Combination of Saffron With Elderberry and Melilotus Blunted Oxidative Damage in ARPE-19 Cells’’ blunted is the not right scientific verb, replace it with suitable other verb.

Response 1: we changed the title as follow: “Combination of Saffron (Crocus Sativus), Elderberry (Sambucus nigra L.) and Melilotus Officinalis protects ARPE-19 cells from oxidative stress”

Comment 2: Introduction:Lines 29-33: The resulting secretion of early hallmarks of inflammation, such as MCP-1, interleukin (IL)-1β and IL-18, activates several intracellular signals, which, in turn, stimulate the production of other cytokines, leading to recruitment of inflammatory cells, thus generating a vicious cycle that rises inflammation and exposes retinal cells to oxidative damages. Running on sentence. Rewrite with proper scientific verbs and remove terminology such as ‘recruitment’. Add the references.

Response 2: we rewrote the period and added the references at the end of the firsty paragraph of the introduction: “ Indeed, oxidative stress impairs retinal cell function leading to the secretion of pro-inflammatory factors, such as Monocyte Chemoattractant Protein-1 (MCP-1), interleukin (IL)-1β and IL-18. These factors trigger several intracellular signaling pathways that stimulate the production of additional cytokines and chemokines [9]. This, in turn, results in an inflammatory environment that activates inflammatory cells, which contribute to exacerbate the inflammatory process, thus creating a vicious cycle that increases both inflammation and oxidative damages to retinal cells [10, 11].”

Comment 3: In introduction, add one paragraph about free radicals responsible for DR and AMD and their mechanism of action.

Response 3: We added a paragraph about free radicals responsible for DR and AMD and their mechanism of action at the start of introduction: “….In addition to oxidative stress resulting from oxygen metabolism, retinal oxidative stress is also associated with the production of Reactive Oxygen Species (ROS) generated by the abundant presence of poly-unsaturated fatty acids (PUFAs) in the outer membrane of the photoreceptors, as well as by the continuous ROS production caused by light exposure [3-5]. Consequently, ROS, such as hydrogen peroxide (H2O2), superoxide and hydroxyl anions, accumulate with aging and their production is further accelerated by hyperglyce-mia [4, 6]. Oxidative stress plays a crucial role in the development of several ocular dis-eases, including age related macular degeneration (AMD) and diabetic retinopathy (RD) [4, 5].

Comment 4: Saffron With Elderberry and Melilotus- what form these components were used? Extract or whole plants powder mixture extract or separately took then mixed together.

Response 4: we added the information about Saffron, Elderberry and Melilotus in the section 5.1. Cell culture and experimental conditions:” Afterwards, cells were pretreated for 24 hours with 40 μg/ml of Saffron (Crocus sativus L.) standardized to contain 3% crocins (Affroneye, Pharmactive Biotech Products, Alcoben-das, Madrid, Spain) (SAF) or various doses (0, 50, 100 and 200 µg/ml) of the high-quality elderberry plant extract Eldosamb® (Sambucus nigra L., Anklam Extrakt GmbH, Anklam, Germany) (E) or of Melilotus Officinalis containing 12% cumarin (Nutraceutica srl, Mon-terenzio, Bologna, Italy) (M) (all the compounds were kindly gifted by VISUfarma spa, Roma, Italy). Then ARPE-19 cells were exposed to 500 µmol/L H2O2 for 3 hours. The opti-mal concentrations for E and M (200 μg/ml) were selected, as was the lowest treatment concentration that improved cell viability of ARPE-19 cells exposed to H2O2. Based on these findings, we used a mix composed of 40 μg/ml Saffron, 200 μg/ml Elderberry, and 200 μg/ml Melilotus for the subsequent experiments. In the next experiments, the growth medium was replaced after 3 days with medium containing Affroneye alone (SAF) or a mix of Affroneye, Eldosamb and Melilotus Officinalis (SEM mix). The next day, 500 µmol/L H2O2 was added to the culture medium for further 3 hours, then cells were processed for each analysis.”

Comment 5: How did the authors decide the concentration of saffron will be effective at 40 μM?

Response 5: Saffron was used at a concentration of 40 µg/ml, as determined by findings from pre-vious studies [Di Paolo et al Efficacy of Hydroponically Cultivated Saffron in the Preservation of Retinal Pigment Epithelium. Molecules 2023, 28, (4).10.3390/molecules28041699; Karimi et al. Crocetin Prevents RPE Cells from Oxidative Stress through Protection of Cellular Metabolic Function and Activation of ERK1/2. International journal of molecular sciences 2020, 21, (8).10.3390/ijms21082949]. Accordingly, we were able to reproduce the same positive results reported in literatures.

Comment 6: Why did authors not use any standard compound that used to treat oxidative stress to compare the results?

Response 6: The aim of this study was to investigate whether the beneficial effects of Saffron may be enhance through combination with other natural compounds. Our focus was not on comparing Saffron to the other compounds. Therefore, we didn’t compare saffron with the other individual components of the mix.

Comment 7: 4.2. Cell viability assays: elaborate the proper procedure with concentration of treatment used for cell viability analysis. Write the formula of cell viability calculations

Response 7: Viable cells were determined using the Cell Titer 96 Aqueous One Solution Cell Prolifera-tion Assay according to the manufacturer's instructions. Absorbances were measured using a spectrophotometer at the wavelength of 490 nm. In each experiment, background absorbance of each cell medium was substracted from those of the relative treatment. Then the results were expressed as the percentage of absorbance in each treatment compared to absorbance value of the control (CTR, 100%). We added this information and the formula of cell viability calculation in Section 5.2. Cell viability assays.

Comment 8: Figure 1C. Arpe-19 cell viability redraw and make uniform ARPE-19 representation. Use the same y-axis intervals as for figure 1a and b used.

Response 8: We redraw figure 1C using the same y-axis intervals as for figure 1A and B, however we maintain a double-segment axis to better highlight the different results obtained with Melilotus.

Comment 9: Figure 6. Representative morphological images of ARPE-19 cells cultured for 24h with S and SEM (upper panel) and then exposed to OX (lower panel). Why did not upper panel cell culture expose together with OX? Does disease occur first or treatment giving before disease occurrence?

Response 9: The upper panel represents cell after 24 hour of culture in standard medium (CTR), and in media containing SAF or SEM mix before exposure to H2O2. The lower panel represents the same batches of cells after exposure of 3 hours to H2O2. We clarified the experimental condition in the figure legend.

Comment 10:  Write the conclusion separately.

Response 10: we wrote the conclusion in a separate section at the end of the Discussion.

Comment 11:  36% plagiarism was found, so reduce the plagiarism to 15%.

Below given sections are plagiarized. Introduction is partial plagiarized while material methods section is mostly plagiarized. Also figure legends were showing plagiarism.

  1. Introduction
  2. Materials and Methods

4.1. Cell culture and experimental conditions

4.2. Cell viability assays

4.3. Caspase-3 activity

4.4. LDH release

4.5. Evaluation of intracellular Reactive Oxygen Species (ROS)

4.6. Measuring of glutathione (GSH) levels

4.7. Caspase-1 activity

4.8. Statistical analysis 

Response 11: we rewrote the majority of the text decreasing the percentage of plagiarisms.

Round 2

Reviewer 1 Report

Comments and Suggestions for Authors

Authors addressed all the Reviewer's comments

Author Response

Comment: Authors addressed all the Reviewer's comments

Response: We sincerely thank you for your valuable suggestions, which contribute to improve the quality of the manuscript.

Reviewer 2 Report

Comments and Suggestions for Authors The authors have made an effort to include additional analysis and revise the manuscript according to the criticisms raised by the reviewers.   The new findings have increased the robustness of the paper, making it suitable for publication. Additional comment: Please provide full plots in the supplementary material.  

Author Response

Comment: The authors have made an effort to include additional analysis and revise the manuscript according to the criticisms raised by the reviewers.   The new findings have increased the robustness of the paper, making it suitable for publication. Additional comment: Please provide full plots in the supplementary material.  

Response: We sincerely thank you for your valuable suggestions, which contribute to improve the quality of the manuscript. We provide a file with the entire original western blot images in the Supplementary Material.